# Active Learning for Semantic Segmentation with Multi-class Label Query

**Sehyun Hwang**    **Sohyun Lee**    **Hoyoung Kim**    **Minhyeon Oh**    **Jungseul Ok**    **Suha Kwak**

Pohang University of Science and Technology (POSTECH), South Korea
{sehyun03, lshig96, cskhy16, minhyeonoh, jungseul.ok, suha.kwak}@postech.ac.kr

## Abstract

This paper proposes a new active learning method for semantic segmentation. The core of our method lies in a new annotation query design. It samples informative local image regions (*e.g.*, superpixels), and for each of such regions, asks an oracle for a multi-hot vector indicating all classes existing in the region. This multi-class labeling strategy is substantially more efficient than existing ones like segmentation, polygon, and even dominant class labeling in terms of annotation time per click. However, it introduces the class ambiguity issue in training as it assigns partial labels (*i.e.*, a set of candidate classes) to individual pixels. We thus propose a new algorithm for learning semantic segmentation while disambiguating the partial labels in two stages. In the first stage, it trains a segmentation model directly with the partial labels through two new loss functions motivated by partial label learning and multiple instance learning. In the second stage, it disambiguates the partial labels by generating pixel-wise pseudo labels, which are used for supervised learning of the model. Equipped with a new acquisition function dedicated to the multi-class labeling, our method outperforms previous work on Cityscapes and PASCAL VOC 2012 while spending less annotation cost. Our code and results are available at https://github.com/sehyun03/MulActSeg.

## 1   Introduction

Supervised learning of deep neural networks has driven significant advances of semantic segmentation for a decade. At the same time, however, it has limited practical applications of the task as it demands as supervision pixel-level class labels that are prohibitively expensive in general. To address this limitation, label-efficient learning approaches such as weakly supervised learning [2, 3, 11, 20, 28, 34, 39, 40, 58, 63], semi-supervised learning [4, 12, 25, 33, 36, 37, 41–44, 48, 49], self-supervised learning [24, 60, 69], and active learning (AL) [9, 10, 14, 23, 31, 38, 47, 55, 56, 68] have been investigated.

This paper studies AL for semantic segmentation, where a training algorithm selects informative samples from training data and asks an oracle to label them on a limited budget. In AL, the design of annotation query, *i.e.*, the granularity of query samples and the annotation format, is of vital importance for maximizing the amount and quality of supervision provided by the oracle within a given budget. Early approaches consider an entire image as a sample and ask for its pixel-wise class labels [56, 68], or select individual pixels and query the oracle for their class labels [54]; they turned out to be suboptimal since the former lacks the diversity of samples [47] and the latter is less budget-efficient as a query provides supervision for only a single pixel.

As a compromise between these two directions, recent AL methods treat non-overlapped local image regions as individual samples [9, 10, 14, 23, 31, 47, 55]. These region-based methods guarantee the diversity of samples by selecting local regions from numerous images with diverse contexts. Also, their queries are designed to obtain region-wise segmentation labels efficiently. For instance, they

37th Conference on Neural Information Processing Systems (NeurIPS 2023).

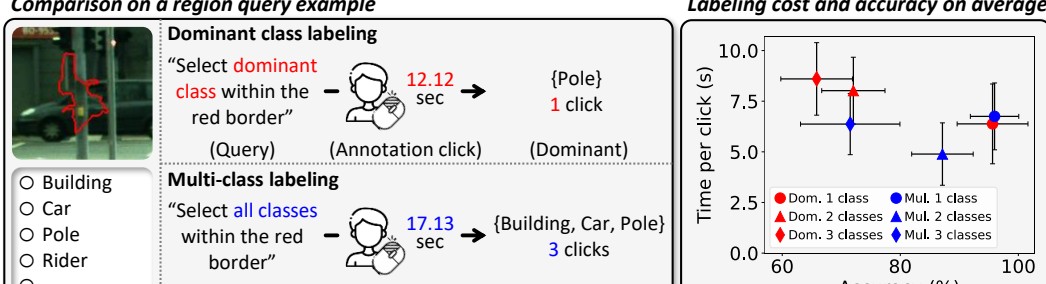

Figure 1: Dominant class labeling [9] versus our multi-class labeling. (*left*) Given a local region as query, an oracle is asked to select the most dominant class by a single click in dominant class labeling, and all existing classes by potentially more than one click in multi-class labeling. As shown here, multi-class labeling often takes less *annotation time per click* because, to determine the dominant one, the oracle has to infer every class in the region after all and sometimes should very carefully investigate the region when the classes occupy areas of similar sizes. (*right*) We conducted a user study to compare the two strategies in terms of actual labeling cost and accuracy versus the number of classes in region queries; the results are summarized in the right plot with one standard deviation. Multi-class labeling took less time per click on average due to the above reason. Furthermore, it resulted in more accurate labels by annotating non-dominant classes ignored in dominant class labeling additionally. Details of this user study are given in Appendix A.

ask the oracle to draw segmentation masks in the form of polygons within an image patch [14, 47], or to estimate the dominant class label of a superpixel so that the label is assigned to the entire superpixel [9]. Although the existing region-based methods have achieved great success, we argue that there is still large room for further improvement in their query designs: Polygon labeling [14, 47] still requires a large number of clicks per query, and dominant class labeling [9] provides wrong supervision for part of a multi-class region. Note that the latter issue cannot be resolved even using superpixels since they frequently violate object boundaries and include multiple classes.

In this context, we first introduce a new query design for region-based AL of semantic segmentation. The essence of our proposal is to ask the oracle for a multi-hot vector that indicates all classes existing in the given region. This *multi-class labeling* strategy enables to prevent annotation errors for local regions capturing multiple classes, and works the same as dominant class labeling (and thus inherits its advantages) for single-class region queries. Moreover, our user study revealed that multi-class labeling demands less annotation time per click and results in more accurate labels compared with dominant class labeling as demonstrated in Fig. 1. However, such region-wise multi-class labels introduce a new challenge in training, known as the *class ambiguity* issue, since they assign *partial labels* [16, 29] (*i.e.*, a set of candidate classes) to individual pixels of the selected regions.

To address the ambiguity issue, we propose a new AL method tailored to learning semantic segmentation with partial labels. Fig. 2 illustrates the overall pipeline of the proposed method. Given a set of local regions and their multi-class labels, our method trains a segmentation network in two stages. In the first stage, the network is trained directly with the region-wise multi-class labels. To this end, we propose two new loss functions for the label disambiguation based on the notions of partial-label learning [16, 29] and multiple instance learning [19], respectively. In the second stage, our method disambiguates the partial labels through pseudo segmentation labels, which are used to train the segmentation network in the supervised learning fashion. To be specific, it finds a set of class prototype features from each local region using the model of the first stage, and employs the prototypes as a region-adaptive classifier to predict pixel-wise pseudo labels within the region. In addition, we propose to propagate the pseudo labels to neighboring local regions to increase the amount of supervision given per query; this strategy benefits by multi-class labeling that enables to propagate pseudo labels of multiple classes, leading to larger expansion of pseudo labels.

Last but not least, we introduce an acquisition function designed to maximize the advantage of multi-class labels in the region-based AL. Our acquisition function considers both uncertainty [32, 62] and class balance [7, 66, 67] of sampled regions so that local regions where the model finds difficult and containing underrepresented classes are chosen more frequently. It shares a similar motivation

with an existing acquisition function [9], but different in that it considers multiple classes of a region and thus better aligns with multi-class labeling.

The proposed framework achieved the state of the art on both Cityscapes [15] and PASCAL VOC 2012 [21]. Especially, it achieved 95% of the fully supervised learning performance on Cityscapes with only 4% of the full labeling cost. In addition, we verified the efficacy and efficiency of multi-class labeling through extensive empirical analyses: Its efficacy was demonstrated by experiments with varying datasets, model architectures, acquisition functions, and budgets, while its efficiency was examined in real-world annotation scenarios by measuring actual labeling time across a large number of human annotators. In short, the main contribution of this paper is five-fold:

- We introduce a new query design for region-based AL in semantic segmentation, which asks the oracle for a multi-hot vector indicating all classes existing within a particular region.

- We propose a novel AL framework that includes two new loss functions effectively utilizing the supervision of multi-class labels and a method for generating pseudo segmentation labels from the multi-class labels, resulting in enhanced supervision.

- To maximize the advantage of multi-class labels, we design an acquisition function that considers multiple classes of a local region when examining its uncertainty and class balance.

- The effectiveness of multi-class labeling was demonstrated through extensive experiments and user study in real-world annotation scenarios.

- The proposed framework achieved the state of the art on both two public benchmarks, Cityscapes and PASCAL VOC 2012, with a significant reduction in annotation cost.

## 2   Related Work

**Active learning (AL).** In AL, a training algorithm samples informative data and asks an oracle to label them on a limited budget so as to maximize performance of a model trained with the labeled data. To this end, AL methods have suggested various sampling criteria such as uncertainty [5, 26, 50], diversity [53, 56], or both [6, 30, 64, 65]. Also, since most of existing AL methods for vision tasks have focused on image classification, the granularity of their annotation queries has been an entire image in general. However, for structured prediction tasks like semantic segmentation, queries should be more carefully designed to optimize cost-effectiveness of annotation.

**Active learning for semantic segmentation.** Most AL methods for semantic segmentation can be categorized into image-based [17, 56, 68] and region-based methods [47]. The image-based methods consider an entire image as the sampling unit and query an oracle for pixel-wise labels of sampled images. These methods have been known to be less cost-effective due to the limited diversity of sampled data; adjacent pixels in an image largely overlap in their receptive fields and thus fail to provide diverse semantics during training. On the other hand, the region-based methods divide each image into non-overlapping local regions, which are considered as individual samples to be selected; As such local regions, image patches [10, 14, 47] and superpixels [9, 35, 55] have been employed. Our paper proposes a new cost-effective region query that allows more accurate and faster annotation, and a new training algorithm taking full advantage of the query design.

**Partial label learning.** Partial label learning [8, 16, 29, 46] is a branch of weakly supervised learning where a set of candidate classes is assigned to each of training data, leading to ambiguous supervision. One primitive yet common approach to partial label learning is to train a model while regarding its top-1 prediction as true labels [8, 46]. However, this approach could be vulnerable to class imbalance [61] and strong correlation between different classes [52]. In contrast, our two-stage training algorithm addresses these issues by disambiguating partial labels by pseudo labeling.

## 3   Proposed Method

We consider an AL process with $R$ rounds. At each round, local regions of a batch are selected using an acquisition function, and then a multi-hot vector (*i.e.*, multi-class label) is assigned to each of them by an oracle. Given the labeled regions, our training algorithm operates in two stages as illustrated in Fig. 2. In the first stage, a segmentation model is trained directly with the region-wise multi-class labels by two loss functions specialized to handle the ambiguity of the labels (Sec. 3.2). In the second

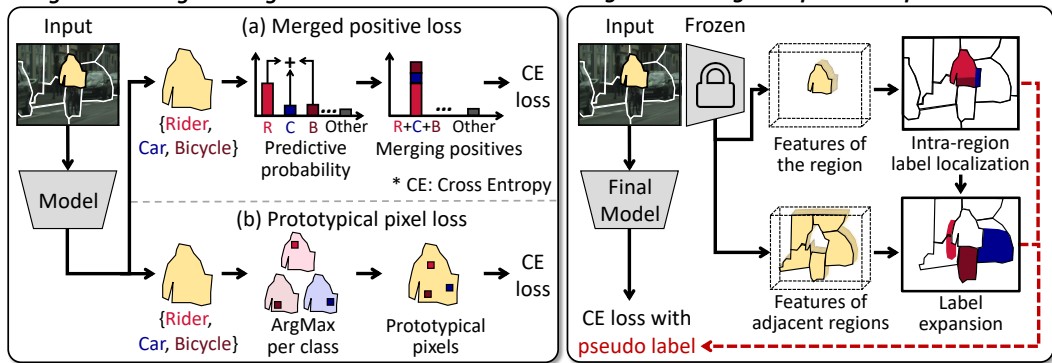

**Stage 1: Learning with region-wise multi-class labels**   **Stage 2: Learning with pixel-wise pseudo labels**

Figure 2: Our two-stage training algorithm using partial labels. (*left*) In the first stage, a model is trained using region-wise multi-class labels through two losses: the merged positive loss that encourages the model to predict any of the annotated classes for each pixel of the region, and the prototypical pixel loss that ensures at least one pixel in the region corresponds to each annotated class. (*right*) The second stage disambiguates the region-wise multi-class labels by generating pixel-wise pseudo labels, which are then used for training the final model. To this end, it first assigns pseudo class labels to individual pixels within the region (*i.e.*, intra-region label localization), and then propagates the pseudo labels to adjacent regions (*i.e.*, label expansion).

stage, the ambiguity is mitigated by generating pixel-wise pseudo labels and using them for training the model further (Sec. 3.3). The remainder of this section describes details of our framework.

## 3.1   Acquisition of region-wise multi-class labels

For an unlabeled image set $\mathcal{I}$, we partition each image $I \in \mathcal{I}$ into a set of non-overlapping regions, denoted by $S(I)$, such that a pixel $x \in I$ belongs to only a unique region $s \in S(I)$. Such a non-overlapping partition can be obtained by a superpixel algorithm as in [9]. Let $\mathcal{S} := \bigcup_{I \in \mathcal{I}} S(I)$ be the set of all the partitions for $\mathcal{I}$. For each round $t$, we issue a batch of regions, denoted by $\mathcal{B}_t \subset \mathcal{S}$, each of which is queried to acquire a multi-class label $Y \subset \{1, 2, \cdots, C\}$, where $|Y| \geq 1$ and $C$ is the number of classes. Then the model $\theta_t$ is trained using the labeled regions obtained so far, denoted as $\mathcal{D} := \bigcup_t \mathcal{D}_t$, where $\mathcal{D}_t$ consists of pairs of region and associated multi-class label, $\mathcal{D}_t := \{(s, Y) : s \in \mathcal{B}_t\}$. The model $\theta_t$ includes a feature extractor $f_t(\cdot)$ and a classifier with a weight matrix $[\mathbf{w}_{t,1}, \mathbf{w}_{t,2}, \cdots, \mathbf{w}_{t,C}] \in \mathbb{R}^{d \times C}$. The predictive probability of pixel $x$ being class $c$ is computed by

$$P_{\theta_t}(y = c | x) = \text{softmax}\left( \frac{f_t(x)^\top \mathbf{w}_{t,c}}{\tau \|f_t(x)\| \|\mathbf{w}_{t,c}\|} \right), \tag{1}$$

where $\tau$ is a temperature term.

**Acquisition function.** We introduce an acquisition function for selecting a batch of regions $\mathcal{B}_t \subset \mathcal{S}$ at round $t$ that aims to optimize the benefits of multi-class labels, while adhering to the principles of previous studies [7, 9, 66] for uncertain and class-balanced region selection. We adopt best-versus-second-best (BvSB) [32, 62] as an uncertainty measure, defined as

$$u_{\theta_t}(x) := \frac{P_{\theta_t}(y = c_{\text{sb}} | x)}{P_{\theta_t}(y = c_{\text{b}} | x)}, \tag{2}$$

where $c_{\text{b}}$ and $c_{\text{sb}}$ are the classes with the largest and second-largest predictive probabilities for $x$ under $\theta_t$, respectively. For class-balanced sampling, we first estimate the label distribution $P_{\theta_t}(y)$ as

$$P_{\theta_t}(y = c) = \frac{1}{|X|} \sum_{x \in X} P_{\theta_t}(y = c | x), \tag{3}$$

where $X := \{x : \exists s \in \mathcal{S}, \, x \in s\}$. Our acquisition function, favoring uncertain regions of rare classes, is defined as

$$a(s; \theta_t) := \frac{1}{|s|} \sum_{x \in s} \frac{u_{\theta_t}(x)}{\left(1 + \nu \, P_{\theta_t}(c_b)\right)^2}, \tag{4}$$

where $\nu$ is a hyperparameter regulating the class balancing effect. Distinct from an existing acquisition function [9] that considers the dominant class only, our function considers classes of all pixels in a region and thus better aligns with multi-class labeling. For the remainder of this section, we will omit the round index $t$ from $\mathcal{D}_t$ and $\theta_t$ for simplicity.

## 3.2 Stage 1: Learning with region-wise multi-class labels

During training of the segmentation model, regions labeled with a single class are used for the conventional supervised learning using the pixel-wise cross-entropy (CE) loss. The set of local regions equipped with single-class labels is defined as

$$\mathcal{D}_{\mathrm{s}} := \left\{ (s, \{c\}) : \exists (s, Y) \in \mathcal{D}, |Y| = 1, c \in Y \right\} . \tag{5}$$

The pixel-wise CE loss is then given by

$$\mathcal{L}_{\mathrm{CE}} = \hat{\mathbb{E}}_{(s, \{c\}) \sim \mathcal{D}_{\mathrm{s}}} \left[ \frac{1}{|s|} \sum_{x \in s} -\log P_\theta(y = c|x) \right] . \tag{6}$$

On the other hand, regions labeled with multiple classes, denoted as $\mathcal{D}_{\mathrm{m}} := \mathcal{D} - \mathcal{D}_{\mathrm{s}}$, cannot be used for training using the pixel-wise CE loss, since a multi-class label lacks precise correspondence between each pixel and class candidates, making it a weak label [16, 19, 29]. To effectively utilize the supervision of $\mathcal{D}_{\mathrm{m}}$, we introduce two loss functions.

**Merged positive loss.** Each pixel in a region is assigned with partial labels [16, 29], *i.e.*, a set of candidate classes. The per-pixel prediction in each region should be one of these candidate classes. This concept is directly incorporated into the merged positive loss, which is defined as

$$\mathcal{L}_{\mathrm{MP}} := \hat{\mathbb{E}}_{(s, Y) \sim \mathcal{D}_{\mathrm{m}}} \left[ \frac{1}{|s|} \sum_{x \in s} -\log \sum_{c \in Y} P_\theta(y = c|x) \right] . \tag{7}$$

This loss encourages to predict any class from the candidate set since the predictive probability of every candidate class is considered as positive.

**Prototypical pixel loss.** Learning with regions assigned with multi-class labels can be considered as an example of multiple instance learning (MIL) [19], where each region is a bag, each pixel in the region is an instance, and at least one pixel in the region must be positive for each candidate class. We call such a pixel *prototypical pixel*, and the pixel with the most confident prediction for each candidate class within the region is chosen as a prototypical pixel:

$$x_{s,c}^* := \max_{x \in s} P_\theta(y = c|x) , \tag{8}$$

where $c \in Y$ and $(s, Y) \in \mathcal{D}_{\mathrm{m}}$. The segmentation model is trained by applying the CE loss to each prototypical pixel with the assumption that the class associated with it is true. To be specific, the loss is defined as

$$\mathcal{L}_{\mathrm{PP}} := \hat{\mathbb{E}}_{(s, Y) \sim \mathcal{D}_{\mathrm{m}}} \left[ \frac{1}{|Y|} \sum_{c \in Y} -\log P_\theta(y = c|x_{s,c}^*) \right] . \tag{9}$$

As reported in the literature of MIL [19], although the prototypical pixels may not always match the ground truth, it is expected that training with numerous prototypical pixels from diverse regions enables the model to grasp the underlying concept of each class. Moreover, this loss mitigates the class imbalance issue as it ensures that every candidate class equally contributes to training via a single prototypical pixel in a region, leading to a balanced class representation.

In summary, the total training loss of the first stage is given by

$$\mathcal{L} = \lambda_{\mathrm{CE}} \mathcal{L}_{\mathrm{CE}} + \lambda_{\mathrm{MP}} \mathcal{L}_{\mathrm{MP}} + \mathcal{L}_{\mathrm{PP}} , \tag{10}$$

where $\lambda_{\mathrm{CE}}$ and $\lambda_{\mathrm{MP}}$ are balancing hyperparameters.

## 3.3 Stage 2: Learning with pixel-wise pseudo labels

In the second stage, we disambiguate the partial labels by generating and exploiting pixel-wise one-hot labels. The pseudo label generation process comprises two steps: *intra-region label localization* that assigns pseudo class labels to individual pixels within each labeled region, and *label expansion*

Figure 3: The pseudo label generation process (*left*) and its qualitative results (*right*). In each of the labeled regions, the feature vector located at the prototypical pixel of an annotated class is considered the prototype of the class, and the set of such prototypes is used as a region-adaptive classifier for pixel-wise pseudo labeling within the region (label localization). The pseudo labels of the region are propagated to adjacent unlabeled regions similarly (label expansion), but for conservative propagation, only relevant pixels that are close to at least one prototype will be assigned pseudo labels.

that spreads the pseudo labels to unlabeled regions adjacent to the labeled one. This process is illustrated in Fig. 3, and described in detail below.

**Intra-region label localization.** For each of the labeled regions, we define a *prototype* for each annotated class as the feature vector located at the prototypical pixel of the class, which is estimated by Eq. (8) using the model of the first stage. The set of such prototypes is then used as a region-adaptive classifier, which is dedicated to pixel-level classification within the region. To be specific, we assign each pixel the class of its nearest prototype in a feature space; the assigned pseudo label for $x \in s$ where $(s, Y) \in \mathcal{D}$, is estimated by

$$\hat{y}(x) := \underset{c \in Y}{\arg\max} \cos\big(f_\theta(x), f_\theta(x^*_{s,c})\big) \;, \tag{11}$$

where $x^*_{s,c}$ is the prototypical pixel of class $c$ and $\cos(f, f') = \frac{f^\top f'}{\|f\|\|f'\|}$ is the cosine similarity between two feature vectors $f$ and $f'$.

**Label expansion.** The rationale behind the label expansion step is that the class composition $Y$ of a region $(s, Y) \in \mathcal{D}$ may provide clues about classes of its adjacent regions $s' \in \mathrm{N}(s)$, where $\mathrm{N}(\cdot)$ denotes a set of unlabeled regions adjacent to $s$, *i.e.*, $\mathrm{N}(s) \cap \mathcal{D} = \emptyset$. Similar to label localization, the label expansion step aims to assign pixels in $\mathrm{N}(s)$ the class labels of their nearest prototypes. However, since $\mathrm{N}(s)$ may contain irrelevant classes, propagating the labels to all pixels in $\mathrm{N}(s)$ could cause incorrect pseudo labels, leading to performance degradation of the final model. Hence, the pseudo labels are proposed only to relevant pixels that are sufficiently close to at least one prototype in the feature space. More specifically, to compute the relevance in a region- and class-adaptive manner, we propose to use prototype-adaptive thresholds: the prototype-adaptive threshold for class $c \in Y$ in $(s, Y) \in \mathcal{D}$ is defined as

$$\alpha_c(s) = \mathrm{med}\Big(\big\{\cos\big(f_\theta(x), f_\theta(x^*_{s,c})\big) : x \in s, \; \hat{y}(x) = c\big\}\Big) \;, \tag{12}$$

where $\mathrm{med}(\cdot)$ yields the median value of a set, $x^*_{s,c}$ is the prototypical pixel of class $c$ (Eq. (8)), and $\hat{y}(x)$ is the pseudo label of $x$ (Eq. (11)). We propagate pseudo labels of the labeled region $s$ in $\mathcal{D}$ to pixels of an adjacent region $\{x : \exists s' \in \mathrm{N}(s), \; x \in s'\}$ by

$$\hat{y}(x) := \underset{c \in \hat{Y}(x)}{\arg\max} \cos\big(f_\theta(x), f_\theta(x^*_{s,c})\big) \quad \text{only if } |\hat{Y}(x)| \geq 1 \;, \tag{13}$$

where $\hat{Y}(x) := \big\{c : \cos\big(f_\theta(x), f_\theta(x^*_{s,c})\big) > \alpha_c(s), c \in Y\big\}$; $x$ is a relevant pixel if $|\hat{Y}(x)| \geq 1$. By using the prototype-adaptive threshold for filtering, we can adjust the amount of label expansion in each region without the need for hand-tuned hyperparameters.

The segmentation model is then further trained using the pixel-wise CE loss with pseudo segmentation labels generated by both of the label localization and expansion steps.

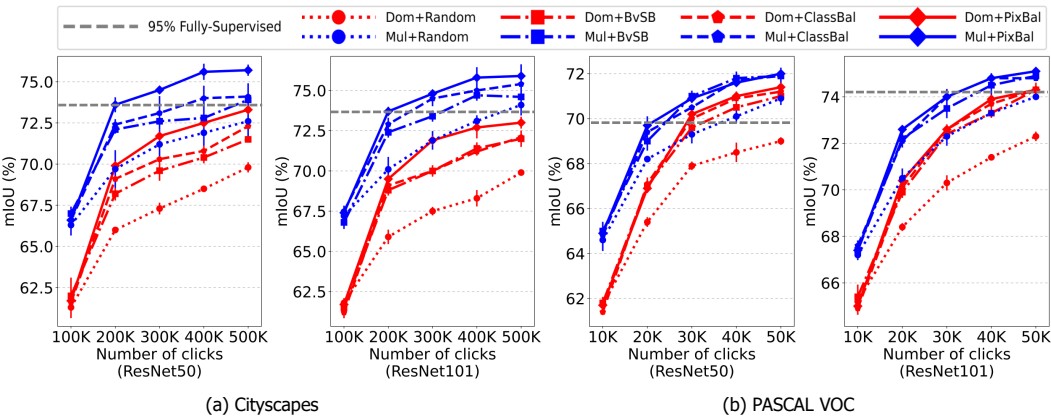

(a) Cityscapes                                    (b) PASCAL VOC

Figure 4: Accuracy in mIoU (%) versus the number of clicks (budget) for dominant class labeling (Dom) [9] and multi-class labeling (Mul) equipped with four different acquisition functions (Random, BvSB, ClassBal, PixBal). The reported accuracy scores are averaged across three trials.

## 4 Experiments

### 4.1 Experimental setup

**Datasets.** Our method is evaluated on two semantic segmentation datasets, Cityscapes [15] and PASCAL VOC 2012 (VOC) [21]. The former contains 2975 training, 500 validation, and 1525 test images with 19 semantic classes. The latter consists of 1464 training and 1449 validation images with 20 semantic classes. We evaluated models on validation splits of these datasets.

**Implementation details.** We adopt DeepLabv3+ [13] with ResNet-50/101 pretrained on ImageNet [18] as our segmentation models, AdamW [45] for optimization. The balancing hyperparameters $\lambda_{CE}$ and $\lambda_{MP}$ of Eq. (10) are set to 16 and 8, respectively, and the temperature $\tau$ was fixed by 0.1. In both datasets we utilize $32 \times 32$ superpixel regions given by SEEDS [59]. For Cityscapes, initial learning rates are set to $2e-3$ (stage 1) and $4e-3$ (stage 2), and $\nu$ in Eq. (4) is set to 6. The models are trained for 80K iterations with mini-batches of four $769 \times 769$ images. We assign an extra *undefined* class for pixels not covered by the original 19 classes. For VOC, we configure $\nu$ to 12 and train the models for 30K iterations using a learning rate of $1e-3$ in both stages. Each mini-batch consists of twelve $513 \times 513$ images. More details are given in the Appendix B.

**Active learning protocol.** Following the previous work [9], we consider the number of clicks as the labeling cost. While this protocol assumes a uniform cost per click, it does not hold in reality as shown in Fig. 1. It is adopted for comparisons with the prior art using dominant class labeling [9], but is *adverse to* the proposed method since our multi-class labeling takes less cost per click than dominant class labeling. We conduct 5 rounds of consecutive data sampling and model updates, with a budget of 100K and 10K clicks per round on Cityscapes and VOC, respectively. The models are evaluated for each round in mean Itersection-over-Union (mIoU) [21] on the validation sets. At the first round, regions are selected at random, and the models are reinitialized with ImageNet pretrained weights per round. We conduct all experiments three times and report the average performance.

**Baseline methods.** We compare our multi-class labeling (Mul) with the dominant class labeling (Dom) in combination with various data selection strategies. Following the established strategies in the previous study [9], we employ Random, which randomly selects superpixels, and the uncertainty-based BvSB given in Eq. (2). ClassBal is BvSB sampling with additional class balancing term proposed in the previous work [9], and PixBal is our sampling method based on Eq. (4).

### 4.2 Experimental results

**Impact of multi-class labeling.** In Fig. 4, we evaluate the performance of multi-class and dominant class labeling across varying budgets, using ResNet50 and ResNet101 backbones, on both Cityscapes and VOC with different acquisition functions. Multi-class labeling constantly outperforms dominant class labeling in every setting across all the compared architectures and datasets. In particular, the

Table 1: The ratio of clicks (%) needed to reach 95% mIoU of the fully-supervised model, relative to full supervision. Results with † are from the prior work [9] using Xception-65 as backbone.

| Query | Sampling | Clicks (%) |
|---|---|---|
| Patch + Polygon | EntropyBox+† [14] | 10.5 |
|  | MetaBox+† [14] | 10.3 |
| Spx + Dominant | ClassBal† [9] | 7.9 |
|  | ClassBal [9] | 9.8 |
|  | PixBal | 7.9 |
| Spx + Multi-class | PixBal (Ours) | **4.0** |

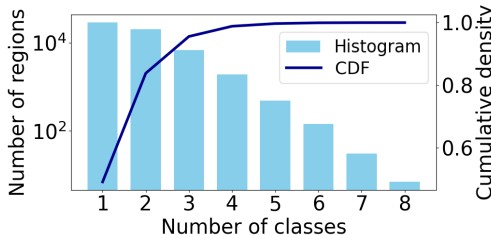

Figure 5: Histogram and cumulative distribution function (CDF) for the number of classes in regions selected at round-5 using PixBal sampling on Cityscapes.

Table 2: Contribution of each component of our method in accuracy (mIoU, %) at each round.

| | Loss function | | Pseudo labeling | | Rnd-1 | Rnd-2 | Rnd-3 | Rnd-4 | Rnd-5 | Avg |
|---|---|---|---|---|---|---|---|---|---|---|
| | $\mathcal{L}_{MP}$ | $\mathcal{L}_{PP}$ | Localization | Expansion | | | | | | |
| (a) | ✓ | ✓ | ✓ | ✓ | 66.6 | 73.6 | 74.5 | 75.6 | 75.7 | $73.2_{\pm0.3}$ |
| (b) | ✓ | ✓ | ✓ | ✗ | 65.2 | 72.2 | 73.5 | 74.3 | 74.7 | $72.0_{\pm0.3}$ |
| (c) | ✓ | ✓ | ✗ | ✗ | 63.8 | 70.3 | 71.8 | 72.6 | 73.3 | $70.4_{\pm0.2}$ |
| (d) | ✓ | ✗ | ✗ | ✗ | 60.8 | 69.7 | 71.0 | 72.3 | 72.3 | $69.2_{\pm0.3}$ |
| (e) | ✗ | ✓ | ✗ | ✗ | 63.1 | 69.9 | 70.9 | 71.2 | 71.9 | $69.4_{\pm0.4}$ |

multi-class labeling model, with just 200K clicks, outperforms the dominant class labeling counterpart that uses 500K clicks on Cityscapes. When using ResNet50, the multi-class labeling model equipped with PixBal sampling, achieves 95% mIoU of the fully-supervised model using only 200K clicks on Cityscapes and 20K clicks on VOC, respectively.

**Impact of the proposed sampling method.** Fig. 4 also demonstrates the efficacy of PixBal. On Cityscapes, PixBal consistently outperforms all the other sampling methods regardless of budget size. It also enhances the performance of dominant class labeling. On VOC, PixBal generally surpasses the baselines, although its improvement over BvSB, which lacks class balancing, is marginal at times since VOC less suffers from class imbalance than Cityscapes. Further analysis on these sampling methods are provided in Appendix C.4.

**Comparison with various query designs.** In Table 1, we evaluate multi-class labeling against baseline methods employing different query designs: drawing polygon mask within an image patch (Patch+Polygon), clicking dominant class within superpixel (Spx+Dominant), and clicking all classes within superpixel (Spx+Multi-class). Following the prior work [9], in this experiment, we measure the ratio of clicks used, relative to the total number of clicks required to draw polygon masks on all images (*i.e.*, full supervision). We then measure the ratio of clicks each method needs to achieve 95% mIoU of the fully-supervised model. As indicated in Table 1, superpixel-based methods typically outperform the baselines using patch or polygon queries. Among these, our multi-class labeling stands out, achieving 95% mIoU of the fully-supervised model using only 4% of its required data.

### 4.3 In-depth analysis on the proposed method

**The number of classes in selected regions.** The histogram and cumulative distribution of Fig. 5 summarize the number of classes within regions selected at round-5 using our PixBal sampling method on Cityscapes. We observe that more than 50% of the selected regions contain two or more classes, explaining the necessity of multi-class labeling. This also suggests that, regarding labeling cost in reality (*i.e.*, actual annotation time), multi-class labeling holds potential for further improvement in efficiency as it requires less labeling time for multi-class regions (Fig. 1).

**Contribution of each component.** Table 2 quantifies the contribution of each component in our method over five rounds: merged positive loss in Eq. (7), prototypical pixel loss in Eq. (9), intra-region label localization, and label expansion. The results show that all components improve performance at every round. The performance gap between (c) and (e) in the table verifies the efficacy of merged

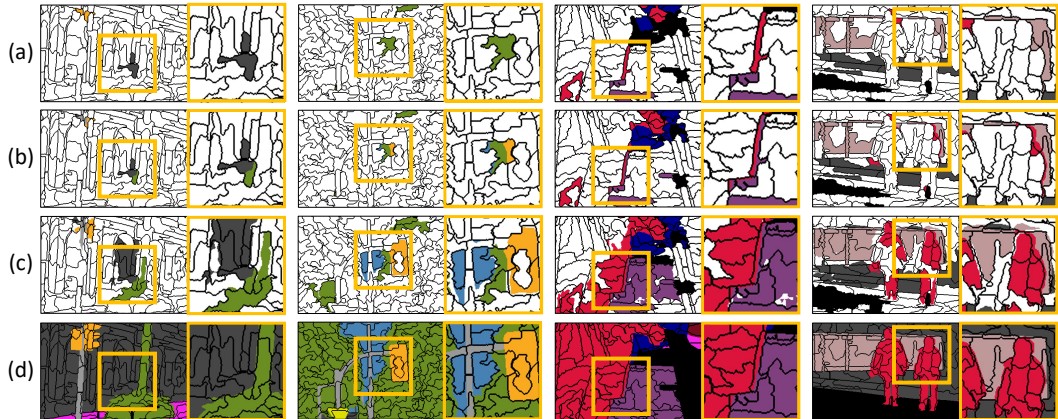

Figure 6: Qualitative comparisons between different labeling strategies. (a) Dominant class labels. (b) Label localization. (c) Label localization + label expansion. (d) Ground-truth.

positive loss for learning with multi-class labels. Meanwhile, the gap between (c) and (d) in the table shows the efficacy of prototypical pixel loss, particularly at the initial round with severe class imbalance due to random sampling. The largest mIoU gain is achieved by intra-region label localization, as shown by the gap between (b) and (c) of the table. Lastly, label expansion further boosts the performance by 1.2%p on average.

**Qualitative analysis on pseudo labels.** Fig. 6 qualitatively compares region-wise dominant class labels and pixel-wise pseudo labels given by the proposed method to show the label disambiguation ability of our method. Regions labeled with dominant classes overlook areas occupied by minor classes. In contrast, our intra-region localization accurately identifies various classes within each region as shown in the second column of Fig. 6. Moreover, label expansion augments the amount of supervision by referencing the class composition within the region. Notably, minor classes within a region often significantly enhance the quality of pseudo labels via label expansion.

**Ablation study of pseudo labeling.** Fig. 7 presents our ablation study on pseudo labeling with ResNet50 on Cityscapes. Fig. 7(a) compares performance improvement by the intra-region label localization (w/ prototypes) and that by a baseline assigning the most confident class among multi-class labels as a pixel-wise pseudo label (w/o prototypes). The result suggests that using prototypes consistently surpasses the baseline across different budgets due to their adaptability to local regions. In Fig. 7(b), we investigate the improvement when label expansion is applied to multi-class and dominant class labels across varying budgets. It turns out that label expansion is

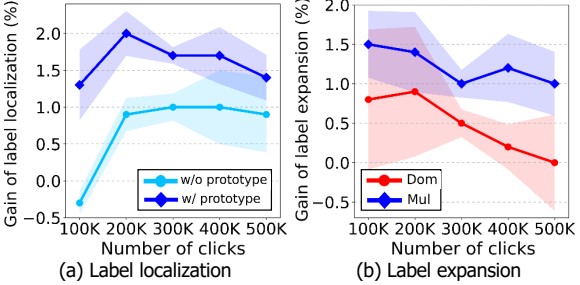

Figure 7: mIoU gain (%) from intra-region label localization and label expansion varying budgets. (a) Gain of label localization with vs. without prototype. (b) Gain of label expansion on dominant and multi-class labels.

more effective with multi-class labeling, as it enables the propagation of pseudo labels belonging to multiple classes, leading to a more extensive expansion of pseudo labels.

**Impact of region generation algorithm.** In Fig. 8(a), we evaluate both dominant class labeling (Dom) and multi-class labeling (Mul) across two superpixel generation algorithms: SLIC [1] and SEEDS [59]. Both labeling methods show better performance when combined with SEEDS. This is because SEEDS is better than SLIC in terms of boundary recall [57] and thus regions generated by SEEDS better preserves the class boundary. Note that the dominant class labeling shows significant performance degradation when combined with SLIC, while the multi-class labeling only shows a modest performance drop. This result suggests that the proposed multi-class labeling is more robust to the quality of the superpixel generation algorithm.

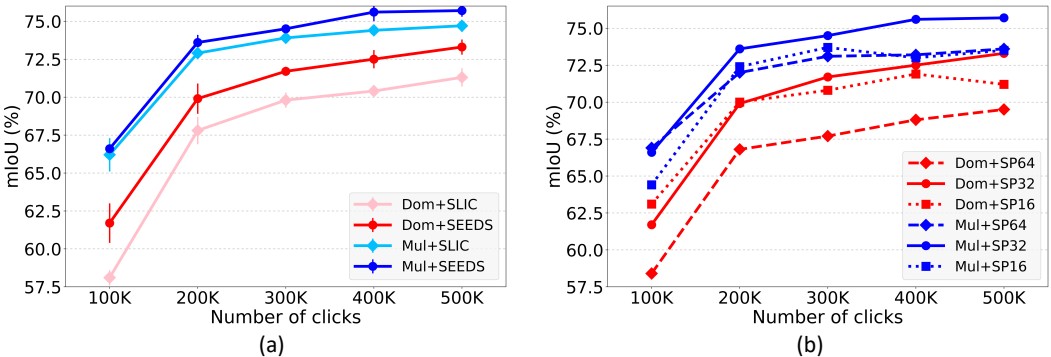

Figure 8: Accuracy in mIoU (%) versus the number of clicks (budget) for dominant class labeling (Dom) and multi-class labeling (Mul) equipped with PixBal evaluated on Cityscapes using ResNet50. (a) Impact of superpixel generation algorithm: SEEDS [59] and SLIC [1]. (b) Impact of superpixel size: 16×16 (SP16), 32×32 (SP32), and 64×64 (SP64).

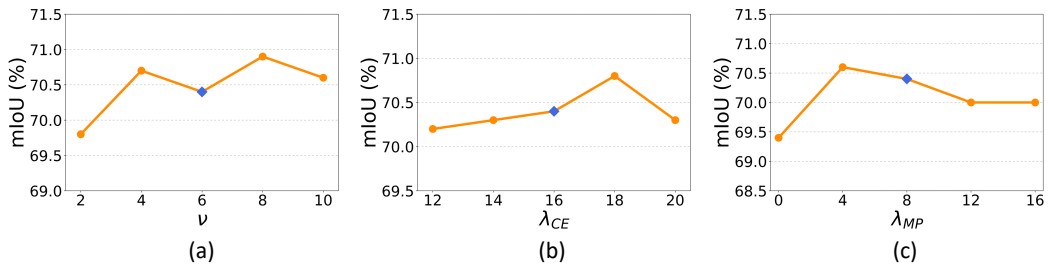

Figure 9: Average accuracy of our stage 1 model over 5 rounds, in mIoU (%), as a function of varying hyperparameters. The model is evaluated on Cityscapes using ResNet50 backbone with PixBal sampling. (a) The class balancing regulation term $\nu$. (b) Loss balancing term $\lambda_{\text{MP}}$. (c) Loss balancing term $\lambda_{\text{CE}}$. The blue diamond marker indicates the value selected for our final model.

**Impact of region size.** In Fig. 8(b), we evaluate dominant class labeling (Dom) and multi-class labeling (Mul) across different superpixel sizes: 16×16 (SP16), 32×32 (SP32), and 64×64 (SP64). Both methods achieve their best with the 32×32 region size. Note that dominant class labeling leads to a large performance drop when the region size increases from 32×32 to 64×64 since a larger region is more likely to contain multiple classes, which violates the key assumption of dominant class labeling. In contrast, multi-class labeling is more robust against such region size variations.

**Impact of hyperparameters.** In Fig. 9, we evaluate the sensitivity of our stage 1 model to variations in the hyperparameters: $\nu$, $\lambda_{\text{MP}}$, and $\lambda_{\text{CE}}$. This evaluation is conducted on Cityscapes using ResNet50 backbone and combined with PixBal sampling. Our model demonstrates robustness to these hyperparameter changes, with accuracy fluctuations of less than 1.5%. It's noteworthy that our final model does not use the optimal hyperparameter values. This indicates that we did not exhaustively tune these parameters using the validation set.

## 5 Conclusion

We have introduced a novel multi-class label query for AL in semantic segmentation. Our two-stage training method and new acquisition function enabled an AL framework with the multi-class label query to achieve the state of the art on two public benchmarks for semantic segmentation by improving label accuracy and reducing labeling cost. Some issues still remain for further exploration. As it stands, our method depends on off-the-shelf superpixels and does not transfer the pseudo labels to the next rounds. Next on our agenda is to address these issues by learning image over-segmentation and developing an advanced AL pipeline.

## Acknowledgments and Disclosure of Funding

We sincerely appreciate Moonjeong Park for fruitful discussions. This work was supported by the NRF grant and the IITP grants funded by Ministry of Science and ICT, Korea (NRF-2021R1A2C3012728, IITP-2020-0-00842, IITP-2022-0-00290, IITP-2021-0-02068, IITP-2019-0-01906, IITP-2021-0-00739).

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

# Active Learning for Semantic Segmentation with Multi-class Label Query
## —*Appendix*—

This appendix provides additional experimental details and findings that have been omitted in the main paper due to the page limit. Sec. A presents a thorough explanation of the user study conducted to compare the cost of dominant class labeling and multi-class labeling. In Sec. B, we explain further details of configurations of our implementation. Sec. C presents additional experiments including the effect of the budget size (Sec. C.1), a comparison with partial label learning loss baselines (Sec. C.2), a comparison with pseudo labeling baselines (Sec. C.3), a comparison with baseline acquisition functions (Sec. C.4), and a qualitative result of our final model (Sec. C.5).

## A  Details of user study

We conducted a user study to compare the dominant class labeling and multi-class labeling in terms of actual labeling cost and accuracy versus the number of classes in region queries. The examples of the questionnaire are illustrated in Fig. 10 and the results are summarized in Table 3. As shown in Fig. 10(a), for each question, annotators received an instruction, an image patch along with a marked local region, and class options. They were requested to select the relevant class options as directed by the instruction. The instructions for dominant class labeling and multi-class labeling were as follows:

> "Select the dominant class that corresponds to the inside of the red boundary.",
> "Select the all classes that exist within the red boundary.".

Prior to the survey, we ensured that every participant reviewed the pre-survey instructional material. This material covered the class composition of Cityscapes, offered the definition of the dominant class and the multi-class labeling, and provided example questions. The pre-survey instructional material is included in the supplementary materials under the name 'pre-survey.pdf'.

As shown in Fig. 10(b), each image patch was a 360-pixel square mostly centered on a local region. Using ground-truth segmentation mask, we divided regions into three groups based on the number of classes (from 1 to 3) present in each region. Twenty regions were then randomly selected from each group for each survey, excluding those containing pixels irrelevant to the original 19 classes, referred to as the 'undefined' class.

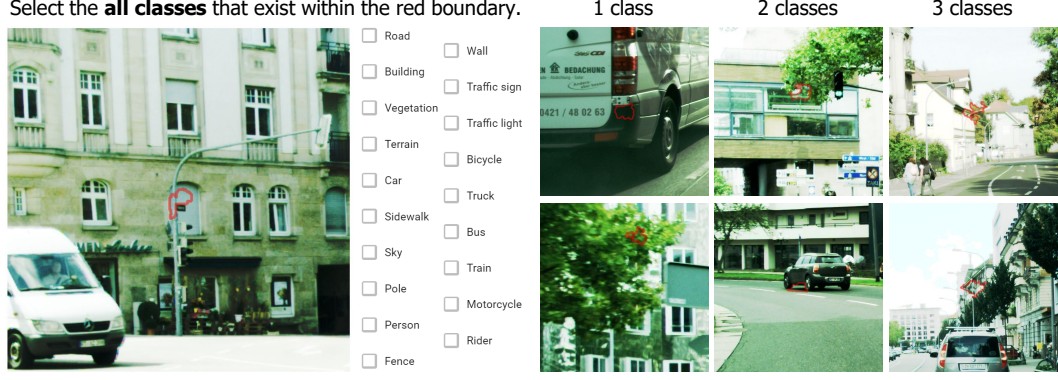

(a) Questionnair for multi-class labeling      (b) Example queries with different number of classes

Figure 10: Questionnaire and local region examples used in the user study. (a) Questionnaire of multi-class labeling survey, consisting of instruction, image patch along with local region marked with red boundary, and class options allowing multiple selections. (b) Examples of local regions used in the user study according to the number of classes present in each region.

Table 3: The result of user study showing the labeling time (second) and accuracy (%) of dominant class labeling and multi-class labeling according to the number of classes within each region.

| Query | # of classes | Total time (s) | Total clicks | Time per click (s) | Accuracy (%) |
|---|---|---|---|---|---|
| Dominant | 1 | $127.6_{\pm 39.4}$ | $20.0_{\pm 0.0}$ | $6.38_{\pm 1.97}$ | $95.63_{\pm 6.00}$ |
| | 2 | $160.5_{\pm 33.0}$ | $20.0_{\pm 0.0}$ | $8.02_{\pm 1.65}$ | $72.05_{\pm 5.36}$ |
| | 3 | $172.1_{\pm 35.8}$ | $20.0_{\pm 0.0}$ | $8.60_{\pm 1.79}$ | $65.83_{\pm 6.07}$ |
| | average | $153.4_{\pm 41.1}$ | $20.0_{\pm 0.0}$ | $7.67_{\pm 2.05}$ | $77.84_{\pm 14.24}$ |
| Multi-class | 1 | $145.5_{\pm 41.8}$ | $21.6_{\pm 2.1}$ | $6.75_{\pm 1.65}$ | $95.97_{\pm 4.10}$ |
| | 2 | $191.6_{\pm 65.8}$ | $39.1_{\pm 3.7}$ | $4.89_{\pm 1.54}$ | $87.14_{\pm 5.21}$ |
| | 3 | $295.8_{\pm 65.3}$ | $49.0_{\pm 8.6}$ | $6.37_{\pm 1.51}$ | $71.52_{\pm 8.42}$ |
| | average | $211.0_{\pm 86.3}$ | $36.5_{\pm 12.5}$ | $6.01_{\pm 1.75}$ | $84.88_{\pm 11.76}$ |

A total of 45 volunteers participated in the survey. We report the results excluding five cases considered outliers in terms of time and accuracy. A unique survey was prepared for each group of regions, categorized by the number of classes. Given three groupings and two labeling methods, a total of six unique forms were prepared. If an annotator annotates the same region twice, there would be a risk of memorizing the image during the first annotation. To avoid this, we asked each participant to answer three out of the six forms, ensuring no region was annotated twice by the same person.

The responses from annotators are evaluated by calculating the Jaccard Similarity (JS) between the ground-truth class set and the responded class set. We define the JS of annotator $u$ as follows:

$$\text{JS}(u) = \frac{1}{|X|} \sum_{i \in X} \frac{|G_i \cap Y_{i,u}|}{|G_i \cup Y_{i,u}|} \ , \tag{14}$$

where $X$ is a set of regions, $G_i$ is ground-truth multi-class label and $Y_{i,u}$ is the set of classes selected by annotator $u$ for region $i$. Note that for dominant class labeling, $|G_i| = |Y_{i,u}| = 1$. We compute the final accuracy as the average JS of all annotators, given by:

$$\text{Accuracy} = \frac{1}{|U|} \sum_{u \in U} \text{JS}(u) \ . \tag{15}$$

As shown in Table 3, multi-class labeling demonstrates comparable efficiency to dominant class labeling for regions with a single class. Moreover, when it comes to regions with multiple classes, multi-class labeling requires less annotation time per click compared to the dominant class labeling.

## B Implementation details

**Configurations.** We implement our method using the PyTorch framework [51]. Following the previous literature [27], we make a slight modification to the original ResNet architecture by replacing the initial 7×7 convolutional layer with two 3× 3 convolutional layers. The output stride of the network is set to 16. We set the learning rate of the backbone to be ten times lower than the standard rate, and we apply a weight decay of $1e-5$. During the training phase, we incorporate several data augmentation techniques, including random scaling ranging from 0.5 to 2.0, random cropping, and random horizontal flipping. To ensure reproducibility and to test the robustness of our approach, we conduct three independent experiments, each initialized with different seed values: 0, 1, and 2.

**Label generation.** Following previous work [9], we assign both dominant class labels and multi-class labels to each region using the ground-truth mask labels. For the dominant class label, we assign the class that dominates the majority of pixels within each region, in accordance with its definition. For the multi-class labels, we attribute all existing classes present within each region. Notably, we disregard classes that appear minimally along the region's boundary during this process. This procedure reflects realistic scenarios where a labeler might fail to recognize classes represented insignificantly on the boundary. More specifically, we implement a binary dilation operation with a 5×5 kernel along the region boundaries, consequently excluding classes that appear on these expanded boundaries. In the user study, we reflect this by marking each local region with a thick, translucent boundary.

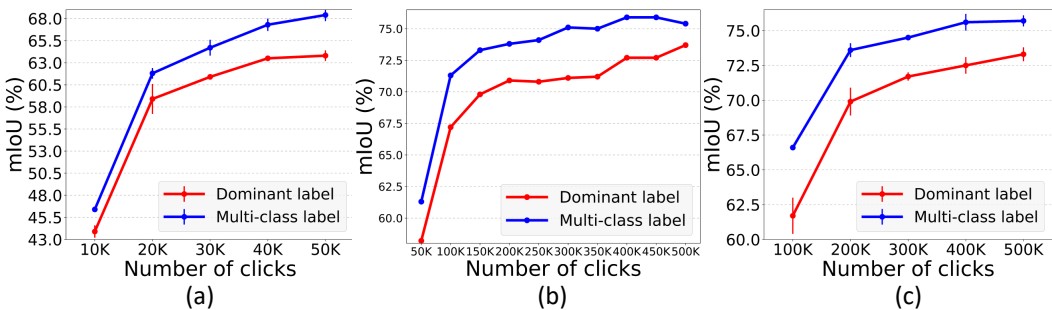

Figure 11: Accuracy in mIoU (%) versus the number of clicks (budget) evaluated on Cityscapes using ResNet50 with PixBal. We evaluate three budget scenarios: (a) 10K clicks over 5 rounds, (b) 50K clicks over 10 rounds, and (c) 100K clicks over 5 rounds.

**Handling undefined class.** In the Cityscapes dataset [15], pixels not covered by the original 19 semantic classes are typically ignored when training segmentation models. On the other hand, in multi-class labeling setting, the precise locations of such uncovered pixels remain unspecified since the multi-class label only provide partial labels. Treating such pixels as belonging to one of the 19 semantic classes naively can misguide the model by providing confusing supervision. Furthermore, active sampling methods like BvsB, ClsBal, PixBal tend to prefer uncertain regions, often leading to the selection of regions containing these uncovered pixels, despite their lack of utility. To address this, we assign an additional *undefined* class for pixels not covered by the initial 19 classes and train the model to predict these undefined classes. As for active sampling, we introduce an extra condition to exclude regions where the predicted dominant class is the undefined class. This undefined class handling strategy is implemented for both dominant class labeling and multi-class labeling.

**Insight for choosing hyperparameters.** We further describe the detailed insights to determine the importance of each loss term; $\lambda_{MP}$, and $\lambda_{CE}$. We roughly determined them by considering the scale of the loss signal back-propagated per pixel. $\mathcal{L}_{PP}$ is applied to only a tiny subset of pixels in an image, *i.e.*, pixels whose number is the same as the number of classes per region (i.e., superpixel). Considering that the final loss value is computed as an average over the pixels, the limited pixel domain of $\mathcal{L}_{PP}$ implies overly stronger loss signals compared with other losses. To counterbalance this side effect, we strategically assigned larger weights to $\lambda_{CE}$ and $\lambda_{MP}$, which are associated with a larger number of pixels. The values of $\lambda_{CE}$ and $\lambda_{MP}$ were estimated by a grid search on the Cityscapes dataset and were used as-is on PASCAL VOC 2012 as well.

## C Additional experiments

### C.1 Impact of budget size

In Fig. 11, we evaluate the accuracy of our final model and dominant class labeling with different budget scenarios: 10K clicks over 5 rounds, 50K clicks over 10 rounds, and 100K clicks over 5 rounds. This analysis is performed on Cityscapes, using a ResNet50 backbone combined with PixBal sampling. As depicted in Fig. 11(a), even in the tiny budget of 10K clicks, the multi-class labeling outperforms the dominant class labeling baseline. This highlights the utility of our method in extremely constrained budget scenarios, where the need for active learning is emphasized. As shown in Fig. 11(b) and (c), when the total budget is kept constant, increasing the frequency of active sampling and model training enhances performance. This improvement can be attributed to the more frequent interactions between the model and the Oracle, leading to more informative active sampling. Multi-class labeling consistently outperforms dominant-class labeling in a wide range of budget sizes, demonstrating the efficacy of our method.

### C.2 Comparison with partial label learning loss baselines

In Fig. 12(a), we compare our proposed losses with baseline partial label learning losses: Infimum loss [8], and RC loss [22, 46]. This comparison is performed on Cityscapes, using a ResNet50 backbone combined with PixBal sampling. As shown in Fig. 12(a), the model with proposed losses

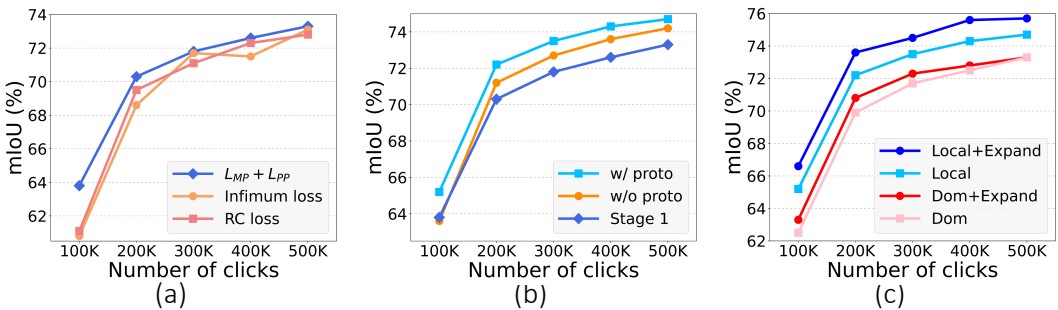

Figure 12: Accuracy in mIoU (%) versus the number of clicks (budget) evaluated on Cityscapes using ResNet50 backbone, with PixBal sampling. (a) The accuracy of our stage 1 model trained with proposed losses ($\mathcal{L}_{MP} + \mathcal{L}_{PP}$), compared with baseline partial label learning losse: Infimum loss [8], and RC loss [22, 46]. (b) The accuracy of stage 1 multi-class labeling model (stage 1), stage 2 multi-class labeling model solely employing intra-region label localization (w/ proto), and a baseline pseudo labeling method without prototype (w/o proto). (c) The accuracy of stage 2 multi-class labeling model with (Local+Expand) and without (Local) label expansion, compared with dominant class labeling model with (Dom+Expand) and without (Dom) label expansion.

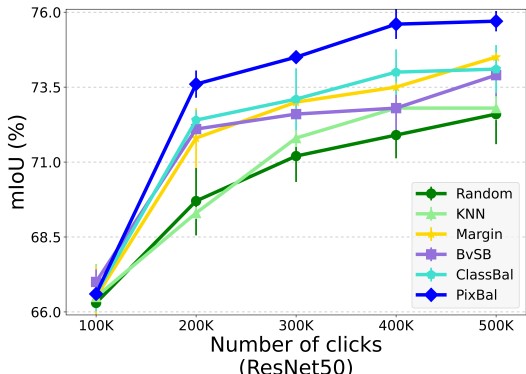

Figure 13: Accuracy in mIoU (%) versus the number of clicks (budget) for multi-class labeling evaluated on Cityscapes equipped with six different acquisition function: Random, KNN, Margin, BvSB, ClassBal, and our sampling strategy (PixBal). The reported accuracy scores are averaged across three trials.

($\mathcal{L}_{MP} + \mathcal{L}_{PP}$) demonstrates superior performance over the models using the baseline losses, particularly in the early rounds.

### C.3 Comparison with pseudo labeling baselines

In Fig. 12(b), we conduct an ablation study comparing our proposed intra-region label localization method (denoted as 'w/ proto') with a baseline intra-region pseudo labeling method that assigns the most confident class among multi-class labels as the pixel-wise pseudo label (denoted as 'w/o proto'). This comparison takes place on the Cityscapes dataset, utilizing a ResNet50 backbone in conjunction with PixBal sampling. While both of the intra-region pseudo-labeling methods improve upon the stage 1 model, the proposed prototype-based label localization demonstrates superior performance over the baseline, specifically in the initial round, where the stage 1 model may lack accuracy.

In Fig. 12(c), we compare the performance improvement brought by label expansion when applied to both the multi-class labeling model (denoted as 'Local+Expand') and the dominant class labeling model (denoted as 'Dom+Expand'). This comparison is also conducted on the Cityscapes dataset, using a ResNet50 backbone paired with PixBal sampling. As shown in Fig. 12(c), label expansion proves to be more beneficial when used with multi-class labeling, as it allows the spread of pseudo labels across multiple classes, resulting in a broader expansion of pseudo labels.

## C.4 Additional comparison with baseline acquisition functions

In Fig. 13, we evaluate multi-class labeling combined with 6 different acquisition functions: Random, K-Nearest Neighbors (KNN), Margin, Best-versus-Second-Best (BvSB), Class Balanced sampling (ClassBal), and our sampling strategy (PixBal). PixBal consistently outperforms all the others regardless of budget size. Margin shows decent performance with a large budget, e.g., it surpasses ClassBal when using 500K clicks. On the other hand, KNN offers modest improvements, marginally surpassing the performance of Random.

## C.5 Qualitative results of our final model

Fig. 14 provides a qualitative result of the predictions of our final model at different rounds. As illustrated in Fig. 14, the quality of the predictions markedly improves as the rounds progress. Notably, the predictions produced by our final model at round 5 exhibit impressive quality, especially when taking into account that it requires only 9.8% of the labeling cost associated with a fully supervised model.

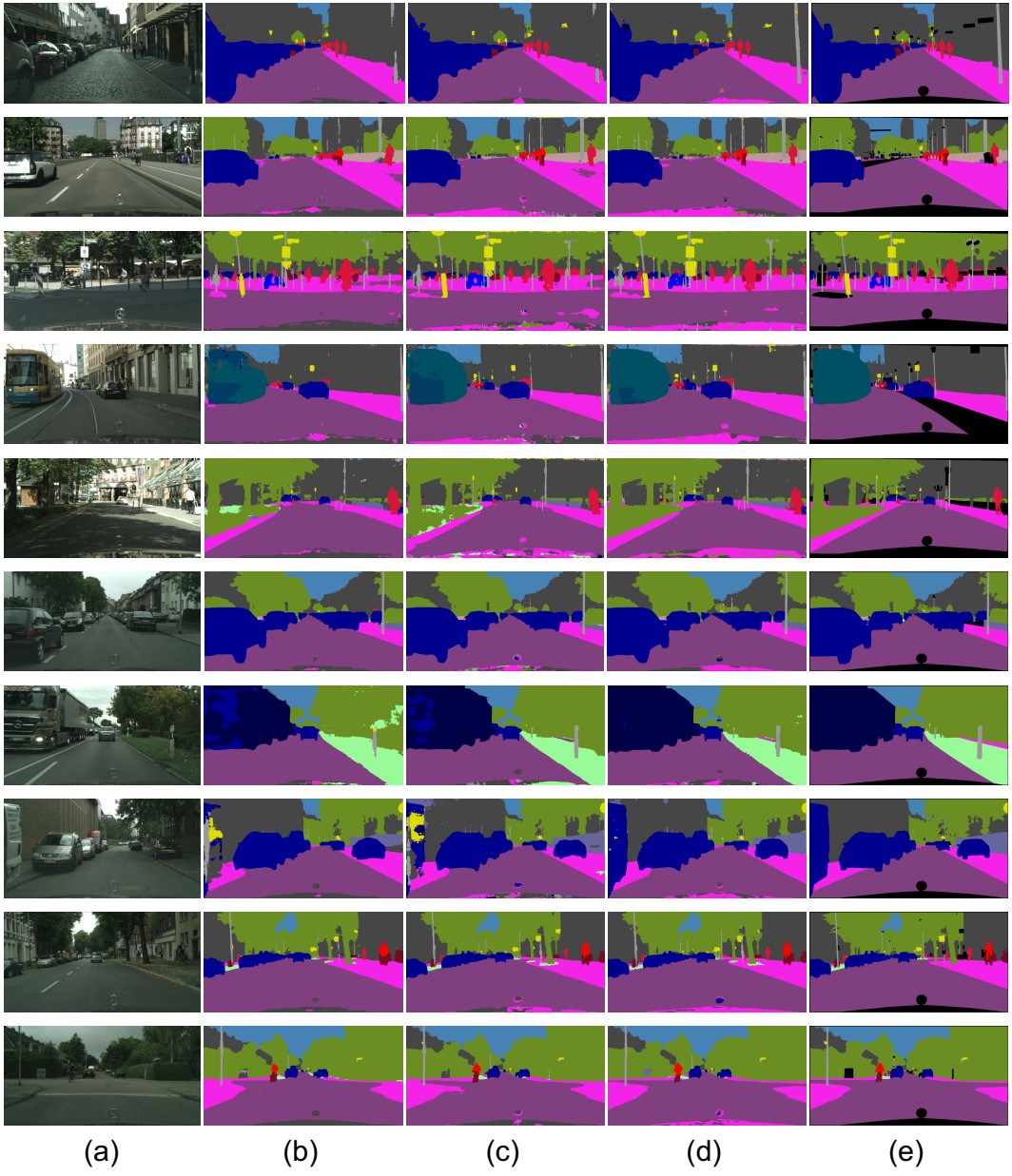

(a)    (b)    (c)    (d)    (e)

Figure 14: Qualitative results of our final model on the Cityscapes dataset. (a) Inputs. (b) Prediction in round 1. (c) Prediction in round 3. (d) Prediction in round 5. (e) Ground Truth.

