# OpenReview forum: "Active Learning for Semantic Segmentation with Multi-class Label Query"
_NeurIPS.cc/2023/Conference — NeurIPS 2023 poster_

### Official Review · Reviewer_FaJw · 2023-06-13

**Soundness:** 3 good
**Presentation:** 4 excellent
**Contribution:** 3 good
**Rating:** 7
**Confidence:** 4

**Summary:**

The paper studied active learning for semantic segmentation. Instead of using each pixel as a query for annotation, this paper used a super-pixel as a query. Since each super-pixel may contain multi-class labels, thus a multi-class labeling scheme is used instead of dominant class labeling. However, it introduced the class ambiguity issue in training since it assigns multi-class labels to each super-pixel region.  The authors proposed a two-step learning method, effectively utilizing the supervision of multi-class labels.  Extensive experiments were conducted to valid the proposed method, where promising results were reported.

**Strengths:**

++ The idea is quite interesting and practical. Using superpixels as active learning query unit is sensible. Assigning multi-class labels to the superpixels is also quite effective with the proposed two-step learning scheme.

++ The paper is well written and easy to follow. I like the figures provided in the paper, they are quite informative.

++ Good performance, clearly performance gain is obtained comparing with that of dominant class labeling.

++  The training scheme includes merged positive loss and prototypical pixel loss, both of which are well designed loss functions. Both are insightful from my perspective.



**Weaknesses:**


--  It would be interesting to see how the two hyperparameters in Eq. (10) affect the performance.

--  It would be better if the number of clicks in Fig.4 can start from 0 instead of 100k. Then we can study the extreme case where very limited number of clicks are provided.


**Questions:**

-- In the right figure of Fig.1,  for the 1-class case, multi-class labeling and dominant class labeling should produce the same accuracy and cost the same amount of time. Why they are slightly different in this figure ?

-- The paper says "we utilize 32 × 32 superpixel regions given by SEEDS". I am wondering how the superpixel generation method and superpixel size affect the performances. This is an important ablation study to conduct.  If the superpixel size is the same as the image size, the method turns to be an image level weakly supervised semantic segmentation task.  If the superpixel size is 1*1, the method is the same as using a pixel as the query unit.

---

> ### Author Rebuttal · Authors · 2023-08-10
>
> We appreciate your insightful feedback and valuable suggestions that helped improve our paper substantially. All the suggestions and experiments will be incorporated into the paper. Please find our responses to the comments below.
> ___
> **Q1. Effect of hyperparameters**
>
> A1. Thank you for the detailed comment. We evaluated the sensitivity of our model to the hyperparameters: $\\nu$, $\\lambda_{MP}$, and $\\lambda_{CE}$ in Figure 2 of the supplementary materials. The results demonstrate that our model is insensitive to these hyperparameters; its accuracy fluctuation was less than 1.5%.
>
> ___
> **Q2. Impact of budget size**
>
> A2. We appreciate your suggestion for an interesting experiment. In Figure 2 of the attached document available at the global response, we evaluate multi-class labeling and dominant class labeling with different budget scenarios: 10K clicks over 5 rounds, 50K clicks over 10 rounds, and 100K clicks over 5 rounds. Multi-class labeling consistently outperforms dominant-class labeling in a wide range of budget sizes, demonstrating the efficacy of our method. Even in the tiny budget of 10K clicks, the multi-class labeling outperforms the dominant class labeling baseline. This highlights the utility of our method in extremely constrained budget scenarios, where the need for active learning is emphasized. These observations will be included in the revision.
>
> ___
> **Q3. Clarifying the results of the user study**
>
> A3. Thank you for the detailed comment. In Table 1 of the supplementary materials, we provide detailed results of the user study illustrated in Figure 1 of the main paper. In a 1-class scenario, the difference in time per click (s) and accuracy (%) between the two labeling methods are 0.37s and 0.34%p, respectively. This difference is marginal when considering the numerical value of the result, which is 6.75s and 95.97% for multi-class labeling. Given that the survey was conducted among 45 participants, such minor variations could exist. Thus, we assume that this slight discrepancy is not significantly meaningful. One possible interpretation for the increased time per click in multi-class labeling is the added time users might need to ensure all classes in the region have been labeled. Examples of the questions from the survey are available in 'pre-survey.pdf' of the supplementary material. We will incorporate this discussion into the paper.
>
> ___
> **Q4. Impact of superpixel generation algorithm and superpixel size**
>
> A4. Thank you for suggesting an important experiment idea!
>
> In Figure 3(a) of the attached document available at the global response, we evaluate both multi-class and dominant class labeling across two superpixel generation algorithms: SLIC [a] and SEEDS [50]. Both of the labeling methods show better performance when combined with SEEDS compared to SLIC. This is likely because SEEDS gives higher boundary recall than SLIC [b], which indicates that the generated region from SEEDS better preserves the class boundary. Note that the dominant class labeling shows significant performance degradation when combined with SLIC, while the multi-class labeling only shows a modest performance drop. This result suggests that the proposed multi-class labeling is more robust to the quality of the superpixel generation algorithm.
>
> In Figure 3(b) of the attached document available at the global response, we evaluate multi-class labeling and dominant class labeling across different superpixel sizes: 16×16 (sp16), 32×32 (sp32), and 64×64 (sp64). Both methods show the best performance at the 32×32 region size. Note that dominant class labeling experiences a significant performance drop when the region size increases from 32x32 to 64x64. This is likely because a larger region is more likely to contain multiple classes, which challenges the assumption of dominant class labeling. In contrast, the multi-class labeling exhibits smaller degradation in the same scenario. This shows that multi-class labeling is more robust to the size of the superpixel.
>
> These experiments will be added to our paper to show additional strengths of our method.
>
> [a] Achanta, Radhakrishna, et al. "SLIC superpixels compared to state-of-the-art superpixel methods." IEEE transactions on pattern analysis and machine intelligence 34.11 (2012): 2274-2282.
>
> [b] Stutz, David, Alexander Hermans, and Bastian Leibe. "Superpixels: An evaluation of the state-of-the-art." Computer Vision and Image Understanding 166 (2018): 1-27.

---

> > ### Comment · Reviewer_FaJw · 2023-08-18
> >
> > Thank you for the detailed response. I have read the review comments from other reviewers and the response provided by the authors. Based on these updates, I maintain my rating as 7.

---

### Official Review · Reviewer_RB6d · 2023-06-19

**Soundness:** 3 good
**Presentation:** 2 fair
**Contribution:** 3 good
**Rating:** 5
**Confidence:** 4

**Summary:**

This manuscript proposes a new issue on active segmentation from multi-class label query. Concretely, the authors are motivated from the shortcomings of existing active learning paradigm, i.e., [1][2] query the whole image pixel annotations, which costs annotation; [3] selects pixels for annotation, which performs with less generalization. To this end, this manuscirpt follows the compromis that actively annotating the uncertain regions with annotating region-level categories and propose a new query design from multi-hot response. To tackle the issues including uncertain region selecting, category ambiguity, and the absence of fine-grained pixel supervision, the manuscript deviced: a region selection strategy (a.k.a.,  acquisition function), training with class-ambiguity loss functions (stage 1 training), self-train learning (stage 2 training). Then, the authors conducted experiments on various benchmarks of semantic segmentation to show their effectiveness.

[1] Samarth Sinha, Sayna Ebrahimi, and Trevor Darrell. Variational adversarial active learning. In Proc. IEEE/CVF International Conference on Computer Vision (ICCV), 2019.
[2] Lin Yang, Yizhe Zhang, Jianxu Chen, Siyuan Zhang, and Danny Z Chen. Suggestive annotation: A deep active learning framework for biomedical image segmentation. In Proc. Medical Image Computing and Computer-Assisted Intervention (MICCAI), 2017.
[3] Gyungin Shin, Weidi Xie, and Samuel Albanie. All you need are a few pixels: semantic segmentation with pixelpick. In Proc. IEEE/CVF International Conference on Computer Vision (ICCV), 2021.

**Strengths:**

1. The draft is beautiful and clear to view. The figures arrayed in Sec. 3 indeed assist me a lot in understanding the detailed methdology and  pipeline.
2. The idea of annotate region-level omni-class is novel. Intuitively, it makes a compromis between image-level with pixel-level as authors claimed in the manuscript.
3. The experiments show that the proposed method is effective in saving annotation budget and promise segmentation performance.

**Weaknesses:**

1. The manuscripts required double-check in some details. For example, in L28, "Early approaches consider an entire image as a sample and ask for its pixel-wise class labels [48, 60], or select individual pixels and query the oracle for their class labels [46]; they turned out to be suboptimal since the former lacks the diversity of samples [40] and the latter is less 30 budget-efficient as a query provides supervision for only a single pixel." It is obvious that indivdual pixel-annotation lacks diversity, while whole image annotation costs heavily. But the author happened to claim the opposite of these two points.
2. Despite that the authors deviced a novel issue in segmenting from region-level multi-class query, the method presented seems to be A+B+C style to me. Moreover, the composition seems not to be novel. Firstly, learning from multi-labels for pre-defined sample-level (i.e., region in this paper, image in learning from multi-class classification) via merging pixel loss is not novel. Secondly, Prototypical Pixel Loss and learning from pseudo labels are very common treatment in self-train paradigm, as cited by authors. Actually, I cannot grab the novelty of the manuscript in methdology design. If I misunderstood some points, it will be better to highlight the contributions and add necessary discussion with previous works.
3. The comparison experiments on region selection (a.k.a., acquisition function) with other classic strategy, e.g., KNN, Margin, et. al are missing. Sample selection is of vital importance for active learning, while the authors seems to narrow down the importance to this part.

**Questions:**

I have no questions in evaluating the manuscript. Code is clear.

**Limitations:**

The authors did not array the limitations. As for the issues proposed in the Sec. Intro., the adopted methods could work intuitively, and experimental results partly support that.

---

> ### Author Rebuttal · Authors · 2023-08-10
>
> We appreciate your insightful feedback and constructive suggestions, which help improve our paper substantially. We will address all the comments and include additional experiments in the revision. Please find our responses to the comments below.
> ___
> **Q1. Clarifying Line 28 in our manuscript**
>
> A1. Thank you for the valuable comment.
>
> Whole image annotation demands labeling every pixel of a selected image, which is extremely expensive. Therefore, only a small number of images can be annotated in general, and accordingly such labeled images are significantly limited in terms of scene diversity.
> On the other hand, pixel annotation aims to label a few pixels of an image. Thus, a substantially larger number of images will be (partly) labeled and accordingly a greater variety of scenes will be covered. However, it is less budget-efficient than the image- and region-based strategies since pixel annotation requires a click per pixel while the image- and region-based counterparts can label a group of pixels (e.g., superpixel or polygon) by a single click.
>
> We will more clearly describe the prior work in the main paper as discussed above and also will proofread the manuscript carefully and thoroughly.
>
> ___
> **Q2. The novelty of our method**
>
> A2. Thank you for the insightful feedback. Our primary contribution is that we are the first to introduce multi-class labeling to active learning for semantic segmentation, as you kindly acknowledged in the 'Strengths' section. The proposed loss and self-training are essential and carefully designed to effectively utilize the multi-class labels. Below, we elaborate on the novelty of our method.
>
> **The proposed loss functions**
>
> First, we would like to emphasize that training a segmentation model with multi-class labels is not straightforward. As these labels lack pixel-wise annotations, the conventional pixel-wise cross-entropy loss for classification is inapplicable. The binary cross-entropy loss commonly used for multi-label classification is also unsuitable here since each pixel is associated with a specific class; this loss assumes the coexistence of multiple labels, which does not hold for individual pixels in semantic segmentation.
>
> Our loss functions resolve this issue by transforming multi-class labels into supervisory signals seamlessly coupled with the standard cross-entropy. For example, the merged positive loss takes the sum of predictive probabilities for candidate classes as positive predictions, and the prototypical pixel loss dynamically selects the most likely pairs of a candidate class and a pixel. These losses have not been employed for active learning to the best of our knowledge, and as reviewer FaJW recognized, we believe these losses could be insightful.
>
> Moreover, our losses are designed to complement each other to effectively utilize multi-class labels. The merged positive loss supplies a weak yet consistent training signal across all pixels in a region, ensuring that any possible noise from the prototypical pixel loss does not misguide the overall training. On the other hand, the prototypical pixel loss gives a stronger signal to a few pixels and is applied uniformly among classes, which helps avoid class imbalance.
>
> Table 1 empirically shows that both losses consistently improve performance across 5 rounds. In Section D.1 and Figure 4(a) of the supplementary material, we also demonstrate the superiority of our losses over baseline partial-label learning losses: infimum loss [7] and RC loss [39].
>
> **The proposed pseudo-labeling approach**
>
> As commented, pseudo labeling is a common treatment in label-efficient learning [2, 10, 20, 24, 32, 41]. However, it is new to use multi-class labels to identify class-representative features within each region and utilize such features as region-adaptive classifiers for precise pseudo labeling.
>
> Figure 7 of the main paper shows that our pseudo-labeling method consistently surpasses the baseline confidence-based approach and is in particular effective when coupled with multi-class labeling.
>
> We will discuss our contributions and compare the proposed method with prior work more clearly and comprehensively in the revision. Thank you for the valuable comment, and we are open to any further suggestions to clarify the contribution of our method.
>
> ___
> **Q3. Additional comparisons with baseline acquisition functions**
>
> A3. We appreciate the valuable suggestion. In Figure 1 of the attached document available at the global response, we evaluate multi-class labeling combined with 6 different acquisition functions: **Random**, K-Nearest Neighbors (**KNN**), **Margin**, Best-versus-Second-Best (**BvSB**), Class Balanced sampling (**ClassBal**), and our sampling strategy (**PixBal**). **PixBal** consistently outperforms all the others regardless of budget size. **Margin** shows decent performance with a large budget, e.g., it surpasses **ClassBal** when using 500K clicks. On the other hand, **KNN** offers modest improvements, marginally surpassing the performance of **Random**. These observations will be included in the revision.
>
> ___
> **Q4. Limitations of our work**
>
> A4. We have stated the limitations and potential future research directions of our study in Lines 300 - 303 of the manuscript, as quoted below, but will present them through a separate paragraph in the revision.
>
> >Some issues still remain for further exploration. As it stands, our method depends on off-the-shelf superpixels and does not transfer the pseudo labels to next rounds. Next on our agenda is to address these issues by learning image over-segmentation and developing an advanced AL pipeline.

---

### Official Review · Reviewer_9EZK · 2023-07-06

**Soundness:** 3 good
**Presentation:** 3 good
**Contribution:** 2 fair
**Rating:** 5
**Confidence:** 5

**Summary:**

This work introduces a new active learning framework for semantic segmentation, involving a novel query design that uses a multi-hot vector for class representation with in a region, two novel loss functions for effective multi-class label supervision, and an acquisition function considering class uncertainty and balance in a local region. This approach optimizes label usage, enhancing supervision and efficiency in semantic segmentation tasks.

**Strengths:**

1) The motivation of this work is clear and interesting. In my opinion, the main strength of this paper lies in the proposed query based on multi-class labels. To be specific, the traditional dominant class labeling suffer from more annotation time per click and bad performance. Therefore, this paper proposed a novel multi-class labeling query and customized two loss and acquisition function for multi-class labeling to handle this problem.
2) The quantitative experimental results are impressive. Only 4% clicks can reach 95% mIoU of the fully-supervised model.
3) Most of this paper are easy to understand.


**Weaknesses:**

- In my view, the Merged Positive Loss and Prototypical Pixel Loss proposed in this paper are essentially minor modifications to the Cross Entropy Loss from a technical standpoint. The Acquisition Function proposed in the paper simply introduces a hyper-parameter to the existing Acquisition Function, incorporating the concept of multi-class.



**Questions:**

- In line 150, Why is Dm represented as D minus Ds? Shouldn't Dm include Ds.
- In Figure 1, the Annotation Click for Dominant Class Labeling and Multi-class Labeling are 12.12s and 17.13s, respectively. The author should cite relevant references to make this data convincing to the reader.
- In Figure 4, we see that all methods based on multi-class labeling perform better than those based on dominant class labeling. In my opinion, this comparison is not quite fair because multi-class labeling inherently provides more information than dominant class labeling, and thus better performance is to be expected.

**Limitations:**

Please see the weakness.

---

> ### Author Rebuttal · Authors · 2023-08-10
>
> We sincerely appreciate your insightful feedback and suggestions that helped improve our paper substantially. All the suggestions will be incorporated into the main text and the appendix. Please find our detailed responses to the comments below.
> ___
> **Q1. Clarification of our contributions from a technical standpoint**
>
> A1. Thank you for the valuable comment. Our primary contribution is that we are the first to introduce multi-class labeling to active learning for semantic segmentation. Also, the proposed loss and acquisition functions are essential and carefully designed for learning with multi-class labels as you kindly mentioned in the 'Strengths' section. Below, we elaborate on the technical contributions of our work.
>
> **The proposed loss functions**
>
> First, we would like to emphasize that the standard cross-entropy loss cannot be used for learning semantic segmentation using region-wise multi-class labels since such a label cannot be converted to pixel-wise annotation.
>
> While the proposed losses are based on the standard cross-entropy as commented, we newly propose to transform multi-class label information into a trainable supervisory signal compatible with cross-entropy (e.g., the merged positive loss takes the sum of predictive probabilities for candidate classes as positive predictions, and the prototypical pixel loss dynamically selects the most likely pairs of a candidate class and a pixel). As reviewer FaJW recognized, we believe these losses could be insightful.
>
> Moreover, our losses are designed to complement each other. The merged positive loss supplies a weak yet consistent training signal across all pixels in a region, ensuring that any possible noise from the prototypical pixel loss does not misguide the overall training. On the other hand, the prototypical pixel loss gives a stronger signal to a few pixels and is applied uniformly among classes, which helps avoid class imbalance.
>
> Table 1 of the main paper empirically shows that both of the two losses consistently enhance segmentation performance across 5 rounds. In Section D.1 and Figure 4(a) of the supplementary material, we also demonstrate the superiority of the losses over baseline partial-label learning losses such as infimum loss and RC loss.
>
> **The proposed acquisition function**
>
> The proposed acquisition function is also aligned with multi-class labels by accounting for all classes present in a region, in contrast to the prior acquisition function (Cai et al., CVPR 2021 [8]) targeting only the dominant class. While it is true that our acquisition function introduces an additional hyper-parameter, denoted as $\nu$, Figure 2(a) of the supplementary material demonstrates that the performance of our method is insensitive to changes in $\nu$.
>
> ___
> **Q2. Clarifying the definition of $D_m$**
>
> A2. We define $D_m$ as the set of regions labeled with multiple classes. Specifically,
> $$  D_m := \\{(s,Y): \exists (s,Y) \in D, |Y| > 1\\}\\;\\;,$$
> where $s$ represents a region, and $Y$ denotes the set of classes present within $s$. Thus, $D_m$ does not contain $D_s$.
> The rationale behind this exclusion is that regions in $D_s$ already have precise pixel-wise annotations, making $L_{MP}$ and $L_{PP}$ unnecessary. We will clarify the definitions of $D_m$ and multi-class labels in the revision.
>
> ___
> **Q3. The annotation time in Figure 1**
>
> A3. Figure 1 of the main paper presents the results of a user study. 12.12s and 17.13s are the actual average times it took for three people to annotate the region on the left-hand side of the figure with dominant and multi-class labels, respectively. Meanwhile, the right-hand side summarizes the average annotation times across all surveyed regions. Details of this survey can be found in Section A of the supplementary material, and example questions of the survey are presented in 'pre-survey.pdf'. We will make sure to include these details in the caption of Figure 1 of the main paper.
>
> ___
> **Q4. Fair comparisons with dominant class labeling in FIgure 4**
>
> A4. We emphasize that the number of clicks is adopted as our budget metric in Figure 4 of the main paper for fair comparisons with dominant class labeling. Please note that a multi-class label involving $n$ classes requires $n$ clicks to be annotated, i.e., given the same number of clicks, multi-class labeling leads to fewer labeled regions compared with those of dominant class labeling. Hence, multi-class labeling does not provide more information given the same number of clicks, and accordingly, the comparisons in Figure 4 are not favorable to multi-class labeling.
>
> Actually, the comparisons are rather favorable to dominant class labeling since it in general demands more time per click as demonstrated in Figure 1 of the main paper. Since annotation time more closely reflects practical labeling cost, our multi-class labeling will demand less human resources than dominant class labeling in practice.
>
> Thank you for the valuable comment and we will discuss the above in the revision.

---

> > ### Comment · Reviewer_9EZK · 2023-08-18
> > **Replying to Rebuttal by Authors**
> >
> > After carefully reading the other reviewers' concerns and the author's responses, I would like to thank the author for addressing all my concerns. I would like to keep the original score.

---

### Official Review · Reviewer_DZFu · 2023-07-26

**Soundness:** 3 good
**Presentation:** 3 good
**Contribution:** 3 good
**Rating:** 6
**Confidence:** 3

**Summary:**

This paper proposes an active learning approach for semantic segmentation using multi-class label queries. Different from dominant class labeling methods, where an oracle is asked to select the most dominant class by a single click, this paper instead designs a multi-class labeling approach that asks the oracle for a multi-hot vector that indicates all classes existing in the given region. Experiments on two benchmarks, Cityscapes and PASCAL VOC 2012, demonstrate promising performance with a significant reduction in annotation cost.

**Strengths:**

1. Reducing annotation costs for segmentation tasks is a long-standing and crucial task. This paper considers active learning to select informative regions from training data and asks an oracle to label them on a limited budget, which is a potential research direction to allow large-scale and annotation-efficient training for segmentation tasks.
2. The proposed method is reasonable and technically sound.
3. The experimental results are comprehensive and verify the effectiveness of the proposed method.
4. The overall paper is well-written and easy to follow.

**Weaknesses:**

1. In stage 1, the total training objective comprises three loss terms. While the sensitivity analysis of the balanced terms $\lambda$ is provided in the supplementary material, I am still wondering whether any findings or insights to determine the importance of each loss term.  More insights or clarifications are encouraged.
2. In Figure 7(a), why does the gain of label localization (%) from the method w/ prototype decrease when the number of clicks increases? More clarifications are suggested.

**Questions:**

As mentioned in Weaknesses.

**Limitations:**

The authors have stated the limitations and provided the future research directions.

---

> ### Author Rebuttal · Authors · 2023-08-10
>
> Thank you for your insightful feedback and constructive comments that help improve our paper substantially. We will incorporate all the suggestions into the paper. Please find our detailed responses to the comments below.
> ___
> **Q1. Insights to determine the importance of each loss term**
>
> A1. We appreciate your detailed comment. We roughly determined the significance of the loss terms by considering the scale of the loss signal back-propagated per pixel. $L_{PP}$ is applied to only a tiny subset of pixels in an image, i.e., pixels whose number is the same as the number of classes per region (i.e., superpixel). Considering that the final loss value is computed as an average over the pixels, the limited pixel domain of $L_{PP}$ implies overly stronger loss signals compared with other losses. To counterbalance this side-effect, we strategically assigned larger weights to $L_{CE}$ and $L_{MP}$, which are associated with a larger number of pixels.
>
> The values of $\lambda_{CE}$ and $\lambda_{MP}$ were estimated by a grid search on the Cityscapes dataset and were used as-is on PASCAL VOC 2012 as well. Figure 2 of the supplementary material demonstrates that the values of $\lambda_{CE}$ and $\lambda_{PP}$ in the main paper are not optimal, which indicates that their values are not exhaustively tuned and that our method is not sensitive to the hyper-parameters.
>
> We will elaborate on the process of determining $\lambda_{CE}$ and $\lambda_{MP}$ in the manuscript.
>
> ___
> **Q2. In Figure 7(a), why does the gain of label localization (%) from the method w/ prototype decrease when the number of clicks increases?**
>
> A2. Thank you for the insightful comment. We first would like to clarify that the values in Figure 7(a) denote relative improvements over the performance of our stage 1 model. Absolute values are in Figure 4(b) of the supplementary material, confirming that our prototype-based pseudo labeling consistently outperformed the baseline across all budgets.
>
> The diminishing performance gains with more clicks in the 'w/ prototype' method can be linked to its rapid convergence due to its high efficacy. This behavior is evident from Figure 4(b) of the supplementary material, where the 'w/ prototype' performance converges rapidly as clicks increase. As models receive more supervision, it is natural for them to converge to the upper bound of supervised learning performance. This holds true for both the 'w/ prototype' and 'w/o prototype' methods. However, the 'w/ prototype' method seems to reach this upper bound faster, probably due to its superior performance over the 'w/o prototype' method, resulting in a reduction in the rate of performance gain as clicks increase.

---

> > ### Comment · Reviewer_DZFu · 2023-08-18
> >
> > Thank you for the detailed response to the reviewers' comments. I have read the review comments from other reviewers and the clarifications provided by the authors. My initial concerns regarding the paper have been addressed satisfactorily. Based on these updates, I maintain my rating of the paper as 6.

---

### Author Rebuttal · Authors · 2023-08-10

We sincerely thank all the reviewers for their constructive comments. All reviewers appreciated the novelty of our multi-class labeling, comprehensive experiments, and the clarity of the paper. In addition, they recognized that our training method is both reasonable and sound (DZFu), well suited to the proposed query design (9EZK), and insightful (FaJw). On the other hand, valuable feedback was also provided: they asked for more discussions on our results (DZFu) and details about our experimental setup (9EZK), questioned the technical novelty of our training method compared to previous work (9EZK, RB6d), and requested more ablation studies (RB6d, FaJw).  We will make every effort to integrate all feedback in the revision.

The attached document includes additional comparisons with classic acquisition functions (RB6d), experiments on extremely low-budget settings (FaJw), and ablation studies on the superpixel size and its generation algorithm (FaJw).

**Potential of multi-class labeling for real applications**

Figure 1 of the main paper shows that multi-class labeling requires less annotation time per click. As multi-class labeling shows superiority over dominant class labeling even under the evaluation settings based on the number of clicks (Figure 4 of the main paper), its advantage will be even more evident if the annotation time per click is further considered for evaluation. Regarding that the total annotation time more closely reflects actual labeling cost, we believe multi-class labeling has greater potential in real-world active learning scenarios.

---

### Decision · Program_Chairs · 2023-09-21

**Decision:**

Accept (poster)

**Comment:**

This paper proposes new active learning method for labeling semantic segmentation dataset. All reviewers recommend to accept this paper and agree that the proposed labeling method could have potentially high impact for the community to build large scale semantic segmentation dataset. Some questions regarding the loss function and hyper-parameters has been addressed through rebuttal. The AC congratulates the authors on the acceptance of their paper and encourage the authors to actually apply the proposed method to create datasets that could benefit the research community.